# Willingness to vaccinate against COVID-19 among Bangladeshi adults: Understanding the strategies to optimize vaccination coverage

Minhazul Abedin[1ʘ], Mohammad Aminul Islam[2ʘ], Farah Naz Rahman[1ʘ], Hasan Mahmud Reza[3,4‡], Mohammad Zakir Hossain[5‡], Mohammad Anwar Hossain[5‡], Adittya Arefin[3‡], Ahmed Hossain[3,6ʘ]*

1 Centre for Injury Prevention and Research, Bangladesh (CIPRB), Dhaka, Bangladesh, 2 Department of Media Studies and Journalism, University of Liberal Arts Bangladesh, Dhaka, Bangladesh, 3 Global Health Institute, North South University, Dhaka, Bangladesh, 4 Department of Pharmaceutical Sciences, North South University, Dhaka, Bangladesh, 5 Hypertension & Research Centre, Rangpur, Bangladesh, 6 Department of Public Health, North South University, Dhaka, Bangladesh

ʘ These authors contributed equally to this work.
‡ These authors also contributed equally to this work.
* ahmed.hossain@utoronto.ca

**Data Availability Statement:** Data are available at https://osf.io/byqzv/.

## Abstract

### Background

Although the approved COVID-19 vaccine has been shown to be safe and effective, mass vaccination in Bangladeshi people remains a challenge. As a vaccination effort, the study provided an empirical evidence on willingness to vaccinate by sociodemographic, clinical and regional differences in Bangladeshi adults.

### Methods

This cross-sectional analysis from a household survey of 3646 adults aged 18 years or older was conducted in 8 districts of Bangladesh, from December 12, 2020, to January 7, 2021. Multinomial regression examined the impact of socio-demographic, clinical and healthcare-related factors on hesitancy and reluctance of vaccination for COVID-19.

### Results

Of the 3646 respondents (2212 men [60.7%]; mean [sd] age, 37.4 [13.9] years), 74.6% reported their willingness to vaccinate against COVID-19 when a safe and effective vaccine is available without a fee, while 8.5% were reluctant to vaccinate. With a minimum fee, 46.5% of the respondents showed intent to vaccinate. Among the respondents, 16.8% reported adequate adherence to health safety regulations, and 35.5% reported high confidence in the country's healthcare system. The COVID-19 vaccine refusal was significantly high in elderly, rural, semi-urban, and slum communities, farmers, day-laborers, homemakers, low-educated group, and those who had low confidence in the country's healthcare

**Funding:** The authors received no specific funding for this work.

**Competing interests:** The authors have declared that no competing interests exist.

system. Also, the prevalence of vaccine hesitancy was high in the elderly population, low-educated group, day-laborers, people with chronic diseases, and people with low confidence in the country's healthcare system.

## Conclusion

A high prevalence of vaccine refusal and hesitancy was observed in rural people and slum dwellers in Bangladesh. The rural community and slum dwellers had a low literacy level, low adherence to health safety regulations and low confidence in healthcare system. The ongoing app-based registration for vaccination increased hesitancy and reluctancy in low-educated group. For rural, semi-urban, and slum people, outreach centers for vaccination can be established to ensure the vaccine's nearby availability and limit associated travel costs. In rural areas, community health workers, valued community-leaders, and non-governmental organizations can be utilized to motivate and educate people for vaccination against COVID-19. Further, emphasis should be given to the elderly and diseased people with tailored health messages and assurance from healthcare professionals. The media may play a responsible role with the vaccine education program and eliminate the social stigma about the vaccination. Finally, vaccination should be continued without a fee and thus Bangladesh's COVID vaccination program can become a model for other low and middle-income countries.

## Introduction

Just a year after the virus was first detected in a Chinese city, the COVID-19 pandemic is an ongoing global threat that has infected over 100 million individuals and caused more than 2 million deaths [1]. To mitigate the effect of COVID-19 by preventing or reducing the transmission, scientific communities around the world are putting their best efforts to develop vaccines against the virus. Looking beyond phase 3 COVID-19 vaccine trials, the findings showed demonstrated effectiveness and thus a few vaccines have been approved by at least one national regulatory body across countries for mass deployment [2]. These are RNA vaccines (from Pfizer-BioNTech and Moderna), conventional inactivated vaccines (from Sinopharm, Bharat Biotech, and Sinovac), and viral vector vaccines (from Gamaleya Research Institute and Oxford-AstraZeneca), which are being used in recent vaccination programs of many countries, including the United States, the United Kingdom, Europe, China, Russia, and India [2].

Bangladesh is also moving forward with its vaccination strategy and inoculation program. The country received the first consignment of the 5 million doses of Covishield vaccines from India under a procurement agreement on January 25, 2021 [3]. Bangladesh has so far purchased 30 million doses from the Serum Institute of India through private Beximco pharmaceuticals under a tripartite agreement. To vaccinate 80% of the total adult population, the vaccination drive was inaugurated on 27th January, and countrywide inoculation commenced on 7th February 2021 [3]. The Bangladesh government published a priority list for the first round of vaccine recipients, including frontline workers and older people aged 40 years and above. A compulsory app-based registration system is developed through which willing individuals from priority list categories will register their interest in vaccination against COVID-19 [4]. The vaccines will be distributed primarily through tertiary healthcare centers in the capital city of Dhaka. Another proportion will be dispersed through district hospitals and Upazila health complexes (1st referral center at primary healthcare level).

Bangladesh aims to implement a five-stage vaccination plan covering 130 million people [3]. To ensure the successful implementation of this countrywide mass vaccination program, it is essential to identify the barriers in the process. Several studies indicate that vaccine hesitancy, acceptance, and refusal would be the biggest challenge for many countries in achieving desired vaccination coverage [5, 6]. In 2019, the World Health Organization (WHO) identified vaccine hesitancy as one of the top ten global health threats. It defined vaccine hesitancy as the reluctance or refusal to vaccinate despite vaccines' availability [7, 8]. Several risk factors associated with vaccine hesitancy and rejection have been reported in studies across different countries focusing on vaccine acceptance; these comprise socio-demographic factors (age, gender, marital status, employment status, income), cost, access to services, safety and effectiveness, level of health literacy, and trust in government, health care systems, and mass media [9–14]. This also applies to the COVID-19 vaccine, as several studies, including a global survey, found variances in vaccine acceptance across different socio-demographic groups [9, 15–17].

Although evidence on encouraging vaccination in general is useful in the context of the current pandemic, COVID-19 vaccine acceptance and uptake pose an enormous challenge [18]. Sentiment that sows doubt and mistrust, as well as the viral dissemination of misinformation, are both leading to a community of vaccine reluctance [19]. Understanding the changing trends of vaccine acceptance over time, which can serve as an early warning system for taking necessary steps to prevent decreases in vaccine trust and acceptance, requires a standard measure of confidence and a baseline for comparison. The study offers a valuable baseline of confidence levels to assess willingness to vaccinate in the context of the COVID-19 pandemic and to help identify where further trust building is required to maximize adoption of new life-saving vaccines.

However, data on vaccine acceptance from Low-and-middle Income Countries (LMIC), mainly from Central and South Asia, is substantially limited [20, 21]. This indicates a need for specific research in these regions to explore the factors that determine the acceptance, hesitancy, and refusal of vaccines, as evidence from developed countries will not be applicable here due to the significant differences in social, cultural, and economic contexts. Therefore, this study aims to identify national trends in the public's intent to take vaccine by sociodemographic, clinical and regional differences among Bangladeshi adults to understand the gap in formulating a comprehensive nation-wide vaccination plan.

## Materials and methods

### Study design and participants

From December 12, 2020, to January 7, 2021, we conducted a cross-sectional survey among adult people in Bangladesh. We conveniently selected eight districts from the six divisions of the country. The eight districts were Dhaka, Chattogram, Jamalpur, Rangpur, Dinajpur, Rajshahi, Khulna, and Pirojpur. The map of the selected districts, including sampling distribution, is given in **S1 Table in S1 File.** Two districts Dhaka and Chattogram, were chosen for data collection from the city. Other six districts were chosen to get data collection either from Upazilas or from the rural areas. The slum people were also chosen from Dhaka city and Chattogram city. With the assumption of equal population size in each of the six districts (except Dhaka and Chattogram), we targeted at least 500 households from each selected district. We also targeted at least 800 households from each of Dhaka city and Chattogram city. This gave us data collection from a target of 4600 households. We targeted more households to be included in our study than the required sample size at 80% power, 95% confidence interval, 50% likely to be vaccinated, and a design effect of 2.

We applied a systematic sampling technique for household selection, and the first household was randomly chosen from the approximate geographical center of an upazila or a village. Data collectors proceeded to the next closest household until required households were sampled from districts. Third, we had a single respondent per household interviewed, preferably the head of the household (18 years and above). The female head of the household or other available adult member of the household was surveyed when the male head of the household was not available. Household members have been described as those who lived for at least one month under the same roof and shared cooking and eating facilities from the same source. We also ensured that the participant lived in the districts for at least two years. The details of the sampling allocation are given in the **S1 Table in S1 File**.

Verbal consents were taken for the study as the participants expressed conviction in signing or giving fingerprints on any paper, and also, many of the people from slums were analphabetic. They were reassured that all the information collected would be kept strictly confidential and would not be used for anything other than research purposes. However, they were provided with a consent paper with detailed contact information of the research investigators for any future query. The Institutional Review Board at North South University, Bangladesh approved the study (2020/OR-NSU/IRB/1003). We included the STROBE statement by completing STROBE Checklist.

## Recruitment and training

The data was obtained and cleaned by a team of 16 enumerators consisting of eight men and eight women. A team of data collectors was then built, including two persons, i.e., one from the districts and another person from the North South University (NSU) graduate students. The interview was held in the language of Bangla. The local data collector asked questions first, and then the member from NSU verified the answer by asking the same question in Bangla. Our two research investigators from North South University arranged a two-day practical training session online about ethics and data collection. The enumerators and data collectors were also briefed about the study objectives, methodology, and questionnaire. The researchers also taught data collectors the techniques of report building and preserving neutrality and well-informed on ethical problems, privacy concerns, cultural awareness, and risk management for COVID infection. Following the training, a pilot study was arranged for the eight study teams and evaluated as a single unit. The aim was to observe the capacity to comprehend the relevant techniques and trouble-some situations while interviewing. We made necessary corrections following the piloting. Afterward, each trained team visited their designated district together to collect the data using a semi-structured questionnaire.

## Data collection

It was made clear to the participants that participation in the study was entirely voluntary. The face-to-face interview took place, one person at a time, to ensure privacy. Since the data collection took place during a pandemic, adequate safety measures were deployed includeding maintaining social distance, wearing a mask, and using hand sanitizers. The respondents were given no monetary or food-item incentives. The questions were read out to the interviewees one by one during the interview and asked which of the scale choices was acceptable. The co-investigators reviewed the data collection sheets for completeness, accuracy, and internal consistency and confirmed them by the principal investigator.

## Independent variables

The independent variables included sociodemographic characteristics, economic condition, underlying disease condition, and infection status for COVID-19 of the participants. The sociodemographic variables included gender, age group (<30, 31–40, 41–50, 51–60, 60 + years), education (no formal education, 1 to 5 years of schooling, 6–10 years of schooling, 11–12 years of schooling, and more than 12 years of schooling), occupation (agriculture, day-laborer, health care workers, monthly paid-jobs, business, housewife, retired, unemployed, student), location of residence (rural, semi-urban, city, urban-slum), and marital status (married, never married, divorced or widowed). Later we categorized retired, unemployed and student participants as not working group.

The economic condition is determined by the family monthly-income of the respondents. The monthly household income was categorized as <10000 BDT (approximately US $130), 10,001–20,000 BDT, 20,001–50,000 BDT, >50,000 BDT.

To collect data for an underlying chronic condition, participants were asked if they were suffering from any of the six common chronic conditions: diabetes, hypertension, chronic kidney disease (CKD), chronic respiratory disease (CRD), chronic heart disease, and cancer. The response was recorded as 'Yes/No'. Further, the participants were asked if they were infected by the SARS-CoV-2 virus, and the responses were recorded as 'Yes/No'. The participants self-reported their chronic illnesses and infections in the survey.

Moreover, participants were asked to share how much confidence they have in the health-care system of Bangladesh. Participant's self-reported response on the level of confidence was categorized as: low, moderate, and high. The participants were also asked about their compliance level with the health safety rules and wear masks during the last seven days. The responses of compliance level were categorized as low, moderate, and high.

## Outcome variable

A close-ended question inquiring 'whether the respondents will be willing to vaccinate against COVID-19 if a safe and effective vaccine is available without cost' was used as the outcome variable. It has three categories of responses: intention to vaccinate, uncertainty to vaccinate, and unwillingness to vaccinate. Intent to vaccinate was considered 'vaccine acceptance' whereas uncertain and unwilling to vaccinate were considered 'vaccine hesitancy' and 'vaccine unacceptance'. Another close-ended question with similar response categories was used, inquiring whether the respondent is willing to pay a minimum of 100 BDT (the US $1.3) for vaccination against COVID-19.

## Statistical analysis

Data were analyzed using R 3.6.2. The questionnaire, R scripts, and data are available at https://osf.io/byqzv/. Intention to vaccination among the respondents was presented using descriptive statistics (frequencies, percentage). Therefore, we performed a cross-tabulation between the outcome measure (Unwillingness, vaccination hesitancy, and vaccine acceptance) and the covariates. A multinomial regression model was fitted to examine the association of socio-demographic variables and overall comorbidity with unwillingness and vaccine hesitancy against COVID-19. The multinomial regression coefficient from the model was exponentiated and are presented as relative risk ratios (RRR) along with corresponding 95% confidence intervals (CI). Here, RRR is defined as the ratio of the probability of an outcome in the exposed group to the probability of an outcome in the unexposed group [22]. As some readers may find odds ratios (OR) easier to interpret than RRR, we included two binary logistic regressions examining i) uncertainty about whether to vaccinate against COVID-19 relative

to those who were very likely to vaccinate and ii) unwillingness to vaccinate compared to those who were very likely to vaccinate in **S2** and **S3 Tables in** S1 File, respectively. We also obtained variance inflation factors (VIF) in the logistic regression models to evaluate potential multicollinearity.

The pattern of missing data in the study sample is presented in Supplement. We found the proportion of missing data ranged from 0.01% for education to 21.5% for the occupation. Also, we did not include 686 missing data for covariates in multivariable analysis.

# Results

## Response rate

Of the 4600 households sampled across the 8 districts, 637 were excluded because the head of the household did not consent. An additional 142 households were excluded because no persons were eligible from the household to be included in the study during the study period. Finally, in our analysis, we had 3646 samples, giving a 79.3% response rate. A thorough calculation of the response rate is given in the **S1 Appendix in** S1 File.

## Characteristics of the respondents

A total of 3646 respondents were interviewed, and their sociodemographic characteristics are given in Table 1. The mean (standard deviation) age of respondents was 37.4 (13.9) years. The majority, 2212 (60.7%), of the 3646 respondents were male, and 1456 (39.9%) of 3645 respondents were under 30 years of age. Also, 1006 (27.6%) of the 3645 respondents were between 31–40 years of age, and another 916 (25.1%) were between the age-group of 41–60 years. Also, 267 (7.3%) of respondents were in the older age category of 60+ years. The educational qualification varied across the respondents, with 1031 (28.4%) of 3627 respondents had 6–10 years of schooling, and 890 (24.5%) of respondents had at least 12 years of schooling. Moreover, 467 (12.9%) of 3627 respondents did not have any formal educational qualification. It appears 1735 (52.4%) of the 3308 respondents were rural residents, and 327 (9.9%) of the respondents were from the semi-urban area. Also, 993 (30%) of the respondents were from city areas, and 253 (7.6%) represented urban slums. The majority, 2779 (76.2%), of the respondents were currently married. Employment varied among the 2863 respondents, with 915 (32%) having a job with a fixed monthly income. About 480 (18%) of the 2863 respondents were housewives or not involved in any formal employment. Moreover, 1366 (37.8%) of the 3618 respondents reported a family monthly-income of less than 10000 BDT (approximately US $130). A family monthly-income of more than 50,000 BDT was reported in 324 (9%) of the 3618 respondents. Also, 1294 (35.5%) of 3641 respondents reported high-confidence in the country's healthcare system. Among 3643 respondents, 612 (16.8%) reported high-compliance with health safety rules and wear a mask during the last 7 days.

## Prevalence of vaccine acceptance

Table 1 presents the prevalence of vaccine acceptance by sociodemographic groups. It appears that 2718 (74.5%) of the 3646 respondents reported their willingness to vaccinate against COVID-19 when a safe and effective vaccine is available without any cost. About 617 (17%) of the respondents were uncertain of their decision to vaccinate at the interview. The remaining 311 (8.5%) did not show a willingness to take a vaccine for COVID-19.

## Sociodemographic and regional differences in vaccine acceptance

Table 1 shows a high proportion (14.6% of 267 respondents) of vaccine unacceptance (unwillingness) among the older (60+) age-group while the proportion was 8.1% in 1456 respondents

**Table 1. Distribution of vaccine acceptance, hesitancy, and unwillingness by socio-demographic groups of the respondents.**

| Variable | Unwilling to vaccinate (n = 311, 8.5%) | Uncertain to vaccinate (n = 617, 17.0%) | Intended to vaccinate (n = 2718, 74.5%) | Total (n = 3646) |
|---|---|---|---|---|
| **Sex (n = 3646)** | | | | |
| Female | 136 (9.5%) | 259 (18.1%) | 1039 (72.5%) | 1434(39.3%) |
| Male | 175 (7.9%) | 358 (16.2%) | 1679 (75.9%) | 2212(60.7%) |
| **Age(years) (n = 3645)** | | | | |
| < = 30 | 118 (8.1%) | 242 (16.6%) | 1096 (75.3%) | 1456 (39.9%) |
| 31–40 | 76 (7.6%) | 170 (16.9%) | 760 (75.5%) | 1006(27.6%) |
| 41–50 | 36 (7.0%) | 66 (12.9%) | 410 (80.1%) | 512 (14.0%) |
| 51–60 | 42 (10.4%) | 74 (18.3%) | 288 (71.3%) | 404 (11.1%) |
| 60+ | 39 (14.6%) | 64 (24.0%) | 164 (61.4%) | 267 (7.3%) |
| **Education (n = 3627)** | | | | |
| No Schooling | 60 (12.8%) | 133 (28.5%) | 274 (58.7%) | 467(12.9%) |
| 1–5 Class | 77 (13.1%) | 120 (20.4%) | 392 (66.6%) | 589(16.2%) |
| 6–10 Class | 70 (6.8%) | 146 (14.2%) | 815 (79.0%) | 1031(28.4%) |
| 11–12 Class | 49 (7.5%) | 101 (15.5%) | 500 (76.9%) | 650(17.9%) |
| >12 Class | 52 (5.8%) | 117 (13.1%) | 721 (81.0%) | 890(24.5%) |
| **Employment (n = 2863)** | | | | |
| Monthly paid-job | 44 (4.8%) | 137 (15.0%) | 734 (80.2%) | 915 (32.0%) |
| Agriculture | 70 (13.8%) | 73 (14.4%) | 363 (71.7%) | 506 (17.7%) |
| Business | 38 (5.9%) | 102 (15.8%) | 505 (78.3%) | 645 (22.5%) |
| Day-labor | 39 (17.2%) | 79 (34.8%) | 109 (48.0%) | 227 (7.9%) |
| Healthcare Worker | 3 (6.4%) | 7 (14.9%) | 37 (78.7%) | 47 (1.6%) |
| Housewife | 51 (10.6%) | 88 (18.3%) | 341 (71.0%) | 480 (16.8%) |
| Not working | 3 (7.0%) | 7 (16.3%) | 33 (76.7%) | 43(1.5%) |
| **Monthly family income (n = 3618)** | | | | |
| < = 10000 | 154 (11.3%) | 267 (19.5%) | 945 (69.2%) | 1366(37.8%) |
| 10001–20000 | 80 (7.6%) | 165 (15.7%) | 804 (76.6%) | 1049(29.0%) |
| 20001–50000 | 44 (5.0%) | 123 (14.0%) | 712 (81.0%) | 879(24.3%) |
| >50000 | 24 (7.4%) | 60 (18.5%) | 240 (74.1%) | 324(9.0%) |
| **Location of residence (n = 3308)** | | | | |
| Metropolitan city | 38 (3.8%) | 146 (14.7%) | 809 (81.5%) | 993 (30.0%) |
| Rural | 202 (11.6%) | 341 (19.7%) | 1192 (68.7%) | 1735 (52.4%) |
| Semi-urban | 28 (8.6%) | 67 (20.5%) | 232 (70.9%) | 327 (9.9%) |
| Slum | 43 (17.0%) | 63 (24.9%) | 147 (58.1%) | 253 (7.6%) |
| **Marital Status (n = 3646)** | | | | |
| Never married | 45 (6.4%) | 94 (13.4%) | 562 (80.2%) | 701 (19.2%) |
| Divorced/Separated/ Widowed | 30 (18.0%) | 35 (21.0%) | 102 (61.1%) | 167 (4.6%) |
| Married | 236 (8.5%) | 488 (17.6%) | 2055 (73.9%) | 2779 (76.2%) |
| **Confidence in the country's healthcare system (n = 3641)** | | | | |
| Low | 58 (9.4%) | 117 (18.9%) | 443 (71.7%) | 618 (17.0%) |
| Moderate | 155 (9.0%) | 298 (17.2%) | 1276 (73.8%) | 1729 (47.5%) |
| High | 95 (7.3%) | 199 (15.4%) | 1000 (77.3%) | 1294 (35.5%) |
| **Self-reported compliance with health safety rules and wear masks (n = 3643)** | | | | |
| Low | 128 (6.5%) | 347 (17.6%) | 1497 (75.9%) | 1972 (54.1%) |
| Moderate | 117 (11.0%) | 156 (14.7%) | 786 (74.2%) | 1059 (29.1%) |
| High | 65 (10.6%) | 113 (18.5%) | 434 (70.9%) | 612 (16.8%) |

aged less than 30 years. Furthermore, intention to vaccinate was lower among day-labors than respondents from other occupations. Only 48% of the 227 day-labors showed intention to vaccinate, while it was more than 70% in respondents from other occupational groups. Vaccine hesitancy (34.8%) and unwillingness (17.2%) were also high in 227 day-laborers. Additionally, respondents with a monthly family-income less than 10000 BDT had the lowest percentage of vaccine acceptance (69.2%) among respondents. Moreover, there is a considerable difference in the percentage of vaccine acceptance between respondents who had formal education and respondents who didn't have any institutional education. Intention to vaccinate was much lower (58.7%) among 467 respondents who had no schooling than the other respondents who had at least one year of schooling exposures. Also, city residents had the highest percentage (81.5% off 993 respondents) of vaccine acceptance, and slum residents (58.1%) had the lowest among geographical locations.

Also, 1729 (47.5%) and 618 (17.0%) of the 3641 respondents shared that they had a moderate and low level of trust in the country's healthcare system of Bangladesh, respectively. An increasing trend of vaccine acceptance appeared with the growing confidence in the country's healthcare system. Furthermore, compliance with the government rules during this pandemic situation was low among the respondents. 1972 (54.1%) of the 3643 respondents reported low compliance with the health safety rules, whereas 612 (16.8%) reported high compliance with the health safety rules for controlling the pandemic. It appears that the proportions of vaccine acceptance by levels of self-reported compliance with healthcare rules were approximately similar.

## Prevalence of vaccine acceptance by self-reported chronic conditions and infection with SARS-CoV-2

The prevalences of vaccine acceptance by COVID-19 infection and self-reported chronic conditions are given in **Table 2**. The table shows that 1171 (32.1%) of the 3643 respondents had any chronic diseases (diabetes, hypertension, CKD, CRD, CHD, or cancer). The respondents who reported chronic diseases showed a lower proportion of vaccine acceptance than those who did not have any chronic conditions. The proportion of vaccine acceptance was 68.7% who reported any of the chronic diseases, while the proportion was 77.4% for respondents without any chronic. The proportion of vaccine unwillingness was also higher for respondents with chronic diseases than respondents without chronic diseases. While stratified by diseases, the highest prevalence of acceptance was found among hypertensive patients (69.8%). The proportion of vaccine unwillingness was high in respondents with CKD (15.6%) and respondents with CRD (14.3%). Both high-hesitancy and high-unwillingness about vaccination were found in the respondents who reported COVID-19 infection.

## Multivariable analysis

The multinomial logistic regression model for predicting the uncertainty and unwillingness to vaccinate against the COVID-19 is shown in **Table 3.** The results suggest that rural people were 1.84 times more unwilling to get a COVID-19 vaccine than the city people (RRR = 1.84, 95% CI = 1.12–3.02). The risk of unwillingness for slum people was 3.79 times more compared to the city people (RRR = 3.79, 95% CI = 1.96–7.34). Gender of people was unrelated to both uncertainty and unwillingness around the COVID-19 vaccine. The people in the age group between 31 and 50 years showed significantly lower uncertainty about the COVID-19 vaccine than people of 18–30. The vaccine's pattern of hesitancy and unwillingness for the younger age group (less than 3o years) and older age group (60+ years) showed similar results.

**Table 2. Distribution of vaccine acceptance, hesitancy, and unwillingness by COVID-19 infection and self-reported chronic conditions.**

| Variable | Unwilling to vaccinate (n = 311, 8.5%) | Uncertain to vaccinate (n = 617, 17.0%) | Intended to vaccinate (n = 2718, 74.5%) | Total (n = 3646) |
|---|---|---|---|---|
| **Diabetes (n = 3643)** | | | | |
| No | 260 (8.2%) | 509 (16.1%) | 2386 (75.6%) | 3155 (86.6%) |
| Yes | 50 (10.2%) | 107 (21.9%) | 331 (67.8%) | 488 (13.4%) |
| **Hypertension (n = 3644)** | | | | |
| No | 262 (8.3%) | 511 (16.3%) | 2365 (75.4%) | 3138 (86.1%) |
| Yes | 48 (9.5%) | 105 (20.8%) | 353 (69.8%) | 506 (13.9%) |
| **Chronic Kidney Disease (CKD) (n = 3644)** | | | | |
| No | 280 (8.1%) | 575 (16.7%) | 2597 (75.2%) | 3452(94.7%) |
| Yes | 30 (15.6%) | 41 (21.4%) | 121 (63.0%) | 192 (5.3%) |
| **Chronic Respiratory Disease (CRD) (n = 3644)** | | | | |
| No | 263 (7.9%) | 547 (16.5%) | 2505 (75.6%) | 3315 (91.0%) |
| Yes | 47 (14.3%) | 69 (21.0%) | 213 (64.7%) | 329 (9.0%) |
| **Chronic Heart Disease (CHD) (n = 3644)** | | | | |
| No | 280 (8.2%) | 552 (16.2%) | 2571 (75.6%) | 3403(93.4%) |
| Yes | 30 (12.4%) | 64 (26.6%) | 147 (61.0%) | 241(6.6%) |
| **Cancer (n = 3644)** | | | | |
| No | 301 (8.5%) | 594 (16.7%) | 2666 (74.9%) | 3561 (97.7%) |
| Yes | 9 (10.8%) | 22 (26.5%) | 52 (62.7%) | 83 (2.3%) |
| **Infected with SARS-CoV-2 (n = 3644)** | | | | |
| No | 281 (8.3%) | 554 (16.4%) | 2544 (75.3%) | 3379(92.7%) |
| Yes | 29 (10.9%) | 62 (23.4%) | 174 (65.7%) | 265 (7.3%) |
| **Self-reported chronic diseases (n = 3643)** | | | | |
| No | 192 (7.8%) | 368 (14.9%) | 1912 (77.4%) | 2472(67.9%) |
| Yes | 118 (10.1%) | 248 (21.2%) | 805 (68.7%) | 1171 (32.1%) |

In terms of the probability of vaccine hesitancy, individuals in the geographic regions were distributed similarly. Compared to the monthly paying working population, farmers (involved with agriculture), day labor, and homemakers showed a significantly higher likelihood of reluctance to vaccinate. The day labor showed probability of vaccine hesitancy was 2.39 times higher than the monthly paid working population (RRR = 2.39, 95% CI = 1.59–3.59). Further, people who had schooling of at least a year were less likely to be unsure or unwilling about a COVID-19 vaccine compared to people without formal education. For example, the people who had schooling of more than 12 years were 65% less likely of vaccine hesitancy compared to people without formal-education (RRR = 0.35, 95% CI = 0.24–0.52)

The probability of being vaccinated is significantly higher for people with comorbidity than without a diseased population (RRR = 1.57, 95% CI = 1.23–2.0). The divorced/ widowed people showed a significantly higher probability of unwillingness than never-married people (RRR = 2.11, 95% CI = 1.01–4.43). The people who had high confidence in the country's healthcare system showed a significantly lower probability of unwillingness about the COVID-19 vaccine (RRR = 0.49, 95% CI = 0.32–0.76) compared to people who had low confidence with the country's healthcare system.

## Vaccine acceptance with a minimum cost

The proportion of individuals who expressed willingness to vaccinate against COVID-19 with an associated cost of BDT 100 (approximately $1.2) was also studied. Of the 3502 respondents, 1627 (46.5%) reported their intention to vaccinate against COVID-19 if a minimum cost is

**Table 3. Associated factors of uncertainty and unwillingness to vaccinate against COVID-19 using a multivariable multinomial regression.**

| | Uncertain to vaccinate | | Unwillingness to vaccinate | |
|---|---|---|---|---|
| | RRR | 95% CI | RRR | 95% CI |
| **Sex (reference: female)** | | | | |
| Male | 0.95 | 0.72–1.25 | 1.24 | 0.83–1.87 |
| **Age group (reference: <30 years)** | | | | |
| 31–40 | **0.72** | **0.55–0.95** | 0.76 | 0.52–1.11 |
| 41–50 | **0.53** | **0.37–0.75** | 0.64 | 0.40–1.03 |
| 51–60 | 0.84 | 0.58–1.21 | 1.05 | 0.64–1.72 |
| 60+ | 1.16 | 0.74–1.80 | 1.35 | 0.76–2.42 |
| **Education (reference: no formal education)** | | | | |
| 1–5 | **0.64** | **0.46–0.90** | 0.96 | 0.61–1.49 |
| 6–10 | **0.40** | **0.29–0.56** | **0.54** | **0.34–0.85** |
| 11–12 | **0.46** | **0.31–0.68** | 0.92 | 0.55–1.55 |
| >12 | **0.35** | **0.24–0.52** | 0.69 | 0.40–1.21 |
| **Location of residence (reference: City)** | | | | |
| Rural | 0.98 | 0.73–1.31 | **1.84** | **1.12–3.02** |
| Semi-urban | 1.37 | 0.91–2.06 | **2.27** | **1.18–4.37** |
| Slum | 1.11 | 0.70–1.74 | **3.79** | **1.96–7.34** |
| **Employment (reference: Monthly paid-job)** | | | | |
| Agriculture | 0.98 | 0.67–1.44 | **3.17** | **1.94–5.16** |
| Business | 0.95 | 0.70–1.29 | 1.27 | 0.79–2.06 |
| Day-labor | **2.39** | **1.59–3.59** | **4.42** | **2.54–7.69** |
| Healthcare Worker | 1.12 | 0.45–2.78 | 1.46 | 0.42–5.12 |
| Housewife | 1.14 | 0.76–1.72 | **2.88** | **1.60–5.16** |
| Not Working | 1.35 | 0.56–3.26 | 1.24 | 0.35–4.44 |
| **Presence of any of the chronic diseases (reference: No)** | | | | |
| Yes | **1.57** | **1.23–2.0** | 1.26 | 0.89–1.77 |
| **Marital status (reference: Unmarried)** | | | | |
| Divorced/ widowed | 1.33 | 0.70–2.53 | **2.11** | **1.01–4.43** |
| Married | 1.36 | 0.94–1.96 | 0.86 | 0.53–1.39 |
| **Confidence in the country's healthcare system (reference: low)** | | | | |
| Moderate | 0.81 | 0.612–1.08 | 0.73 | 0.50–1.07 |
| High | **0.68** | **0.49–0.92** | **0.49** | **0.32–0.76** |

Note: Bold faces show significant at 5% significance level. Intention to be vaccinate was the reference group in the multinomial regression model.

involved in vaccination. Barplots of the proportions of the minimum fee of vaccine acceptance by occupational groups and residence place is provided in the **S1 and S2 Figs in S1 File,** respectively. It appears that the proportion of vaccine acceptance with a minimum cost decreased to 20.4% among day laborers, which was 48% with a free vaccination. On the other hand, the percentage of vaccine acceptance with a cost decreased to 73.2% for health care workers, which was 78.7% with a free vaccination strategy. The figure also shows that day laborers, farmers, and homemakers could neglect vaccination even if a minimum cost is involved in the vaccination strategy. If immunization is applied at a minimal rate, the interest of vaccination for slum and rural communities will be reduced to 22.5% and 37.2%, respectively.

## Bangladesh's vaccination coverage until March 25, 2021

Bangladesh began COVID-19 mass vaccination campaign on February 7th, and as of March 25th, 2021, approximately 5 million people had received their first dose of the vaccine [23]. However, substantial disparities have been noticed regarding enthusiasm for vaccination among people from different sociodemographic contexts. The number of females administered the vaccine is almost half in comparison to the number of males taking the first dose [23]. There is a considerable low number of registration from rural communities in comparison to the number of registration from urban areas [24, 25]. The number of registration from villagers is found less than half compared to people from city corporations in some places [25]. The lack of registration for vaccination also extends to urban slum dwellers who have historically been unaware of COVID-19 risk perception and mitigation steps [26]. These slum dwellers tend to be uninterested about getting the vaccine, which may be due to a lack of smartphones with internet access and understanding of how to register on the website. Since a large percentage of rural citizens and all slum dwellers are from low socioeconomic and educational backgrounds, existing policies must be revised to include an extensive awareness campaign in these areas as well as an on-site registration system as an alternative to the app-based registration system [27].

## Discussion

To the best of our knowledge, this is the first study on vaccine acceptance in Bangladesh, which extensively described the predictors and factors influencing uncertainty and unwillingness to vaccinate against COVID-19 as distinct outcomes. The study revealed that 74.5% of Bangladeshi adults had the intention to vaccinate against COVID-19. However, about one-fourth of the study participants were unwilling or uncertain to accept a vaccine for coronavirus. Our prevalence of vaccine acceptance was similar to a population-based study conducted in France, Denmark, Australia, Mexico, India, and Ireland [6, 15, 17, 20, 21]. Though few other studies conducted in the UK (63.5%-67%), Kuwait (23.6%), Saudia Arabia (64.7%), Russia (54.9%), and Italy (53.7%) showed a lower prevalence of vaccine acceptance compared to our study [18, 28–32]. Other studies from China (91.3%), Malaysia (94.3%), Indonesia (93.3%), Ecuador (97.0%), and Brazil (85.4%) showed a higher prevalence of COVID-19 vaccine acceptance [14, 33–35]. As the coronavirus is a novel disease, there may be mistrust and negative beliefs regarding the vaccine, lack of trust in the existing healthcare system, and information gaps, which is very common in an LMIC country setting like Bangladesh [36].

The study revealed location of residence, occupation, marital status, presence of chronic condition and confidence in the country's healthcare system as the significant associated factors for vaccine unwillingness among the Bangladeshi population. Unwillingness or hesitancy for vaccine differed significantly across geographical locations where residents of slum, semi-urban and rural areas were found to be more resistant to vaccine-acceptance than city residents. Almost 40% of the slum dwellers were either hesitant or unwilling to vaccinate against COVID-19. Bangladesh has an estimated four million slum dwellers living in the capital city Dhaka [37]. They are a socio-economically disadvantaged group and lack knowledge regarding COVID-19 and have inadequate preventive measures against the virus infection [38]. This possibly can contribute to their negative attitude toward unwillingness to vaccine. Close results have been found in a similar survey in Mumbai slum in India, which also demonstrated a 20% unacceptance of vaccine for COVID-19 among the slum dwellers [39].

Furthermore, 3 in every 10 study participants from the rural and semi-urban areas were either uncertain regarding vaccination against COVID-19. Such uncertainty was three times higher compared to the urban residents. According to a nationwide poll conducted in the

United States, rural residents had a higher rate of vaccine hesitancy [40]. Similarly, a cross-sectional study in China also revealed a lower acceptance rate for the COVID-19 vaccine among the participants from rural areas [41]. Rural regions of Bangladesh have a lower literacy rate [42]. A couple of studies among the Bangladeshi population also reported a low level of knowledge and preventive practice for COVID-19 in these regions compared to urban areas [35, 43]. The comparative low-risk perception for COVID-19 and low level of education explains the lower acceptance rate for the vaccine in this region compared to the vaccine acceptance in city areas. This demonstrates a need for tailored vaccine deployment strategies for populations across different geographical regions.

The study also revealed varied prevalences of vaccine hesitancy and unwillingness across occupations. Occupations such as agriculture, day-labor, and homemakers showed a low prevalence of vaccine acceptance. Day-labors reported the highest reluctance among the occupation categories represented in our study. Strong unwillingness was observed compared to persons who had a monthly paid-job during this pandemic in Bangladesh. Besides, farmers (agricultural workers) contributed to the refusal in this study. In terms of literacy, the rate is low in both agricultural workers and the day-labor group. These particular two vulnerable groups, in general, fall into the lowest wealth index category lacks awareness [44]. Farmers mostly reside in rural areas, and a recent study suggested that compared to urban, rural residents were particularly at risk of COVID-19 due to their significantly lower level of knowledge [45]. Additionally, they perceive the severity of the pandemic lightly [38]. Maintaining social distance is challenging due to the working nature of the daily wage earners as these working-class take public transport and live in the slums. It is unlikely that the attitude towards the pandemic will be satisfactory as recommended by WHO, which is rather fancy and impractical for slum dwellers [46]. Homemakers (housewives), on the other hand, reported a high level of resistance to being vaccinated against COVID-19. The decision-making domain in a household may have an impact on the women's willingness to vaccinated. The household head in Bangladeshi culture is a male, and women's lack of decision-making autonomy can be related to their unwillingness to get vaccinated against COVID-19 [47].

Moreover, our results showed that about one-third of the respondents with chronic diseases were uncertain or unwilling to take a vaccine against COVID-19. Low level of awareness, concern about effectiveness and its potential side effects, and lack of trust in vaccines are factors that might contribute to the unwillingness. Research from a highly infected country like Italy found that patients with chronic diseases were highly concerned about the vaccine's side effects. They were less willing to take a vaccine than other respondent groups [48]. A population-based study conducted in Germany found that people with chronic diseases were not vaccinated against flu due to mistrust of the vaccination and low perception of the disease's risk [49]. This has serious implication as adverse consequences of COVID-19 is prevailing among individual with chronic conditions. Additionally, as a significant portion of individuals with a chronic illness were uncertain, awareness through media and healthcare workers can motivate and substantially increase the prevalence of vaccine acceptance among this group.

Marital status was revealed to be another significant factor for COVID-19 vaccine acceptance. Respondents who were divorced, separated, or widowed were found twice more likely to be unwilling to get a vaccine than single or unmarried. Another nationwide web-based survey in Saudi-Arabia found an association between marital status and COVID-19 vaccine acceptance [50], where married individuals were found more interested in a vaccine. Perceived risks and attitudes for a disease condition may differ by relationship status, which subsequently shapes the decision for undertaking a vaccine [51].

Demographically, vaccine intention was almost similar in both male and female respondents in our study. The rate of refusal to take vaccine among females was higher than male

respondents. Although sex was not a significant predictor of vaccine acceptance in our study, male sex has been identified as a common positive predictor in many studies worldwide [6, 41].

Furthermore, although statistically not significant, there was a substantial difference in percentages of vaccine acceptance among age groups, income, education, between respondents with a chronic condition and without a chronic illness, and between who were previously infected with COVID-19 and respondents who are still uninfected. The rate of unwillingness to take a COVID-19 vaccine was almost double among the 60+ years population compared to other age groups. The percentage of vaccine hesitancy was also highest among the older group. In total, a little more than one-third of the more aging population of this study held a negative attitude towards vaccination for COVID-19, which is concerning as these groups are the most vulnerable to the adverse outcome of coronavirus infection. The older population of Bangladesh is also substantially lagged in literacy rate than other age-groups [52], which can mold their perception and knowledge regarding COVID-19 and thus influence the decision for vaccination. This differs from the vaccine acceptance rate among the older population of the US and Saudi-Arabia, who found a higher prevalence of acceptance in this group [29, 53]. This unfolds a scenario specific to the LMIC context and demands special attention to this vulnerable group.

Furthermore, the respondents' education and income increased the percentage of COVID-19 vaccine acceptance with growing years of schooling and family income. This is consistent with the findings from a global vaccine acceptance survey involving 19 countries where respondents with high income and higher education were more likely to vaccinate against COVID-19 [19]. The knowledge, attitude, and practice surveys in Bangladesh regarding COVID-19 also revealed higher income and education as determining factors for higher knowledge and preventive practice [29, 36], which explains the high acceptance rates for vaccines in these groups. Also, individuals infected with COVID-19 previously were unwilling to vaccinate than respondents who haven't contracted the virus. This indicates a lack of health communication as there is a common misbelief that a person achieves immunity if the individual is recovered from the COVID-19 infection, which is possibly contributed to the lack of willingness among this group.

The prevalence of a COVID-19 vaccine acceptance among the respondents steeped down strikingly when there is a vaccine cost. Although the Bangladesh government is continuing to distribute vaccines without a fee, emphasis should be given to associated costs such as travel costs, workday loss, etc. While day laborers and people with low income are already less willing for vaccination, any associated cost can potentially increase the chance of vaccine hesitancy or unwillingness among them.

## Strength and limitation

The study has few limitations. The findings are a snapshot taken at a point in time before the rollout of vaccination. Still, with frequent changes in the perceived risk of disease and the development of COVID-19 vaccines themselves, individuals may change. Although this study comprehensively explored the sociodemographic determinants of vaccine acceptance, the influence of essential factors like misinformation on vaccine safety and effectiveness on the intention to vaccinate was not explored in this study. Relationship with trust in the various sources of information such as healthcare sectors and media with vaccine acceptance has also not been addressed, which could also increase the study's strength. Howevver, this is the first study in Bangladesh that has explored the prevalence of vaccine acceptance along with its sociodemographic determinants. To the best of the authors' knowledge, it is also the first study

in an LMIC and one of the few globally used face-to-face data collection regarding COVID-19 vaccine acceptance and thus omits the limitations of online surveys. It contributes to the evidence base for vaccine acceptance in the LMIC context and has significant policy implications for resource-constrained settings like Bangladesh.

## Conclusions

The results of the analysis illustrate the challenges in introducing a new treatment protocol or some other health initiative through a population in a developing country like Bangladesh. The findings will aid policymakers in developing successful immunization policies, implementing vaccination programs, locating and prioritizing accurate target populations to ensure easy access to COVID-19 vaccines, addressing vaccine hesitancy concerns, and increasing public interest in the vaccine. Media and communication experts will be benefitted from crafting their message targeting the right audiences effectively. Our findings highlight that the Government of Bangladesh should take effective steps to develop a tailor-made vaccine campaign strategy for rural people and slum dwellers, especially targeting farmers, day laborers, and homemakers. Communication messages must be designed to be interpreted easily for people with low literacy; thereby, these people can grow trust in the healthcare system and accept the COVID-19 vaccine. The government must take the necessary steps to ensure the convenience of vaccination in rural areas and the accessibility for the slum dwellers. Attention should also be given to individuals suffering from chronic illness with personalized health messages from healthcare professionals.

## Supporting information

**S1 File.**
(DOCX)

## Acknowledgments

All authors acknowledge the graduate students from North South University who were involved in data collection. We would also like to thank the two anonymous reviewers and the editor for insightful comments that improved the presentation and clarity of our manuscript.

## Author Contributions

**Conceptualization:** Minhazul Abedin, Mohammad Aminul Islam, Farah Naz Rahman, Hasan Mahmud Reza, Mohammad Zakir Hossain, Adittya Arefin, Ahmed Hossain.

**Data curation:** Mohammad Zakir Hossain, Mohammad Anwar Hossain, Adittya Arefin, Ahmed Hossain.

**Formal analysis:** Ahmed Hossain.

**Investigation:** Farah Naz Rahman, Hasan Mahmud Reza, Ahmed Hossain.

**Methodology:** Ahmed Hossain.

**Project administration:** Minhazul Abedin, Mohammad Aminul Islam, Farah Naz Rahman, Ahmed Hossain.

**Resources:** Minhazul Abedin, Hasan Mahmud Reza, Mohammad Zakir Hossain, Mohammad Anwar Hossain.

**Software:** Ahmed Hossain.

**Supervision:** Ahmed Hossain.

**Writing – original draft:** Minhazul Abedin, Mohammad Aminul Islam, Farah Naz Rahman, Hasan Mahmud Reza, Ahmed Hossain.

**Writing – review & editing:** Minhazul Abedin, Mohammad Aminul Islam, Farah Naz Rahman, Hasan Mahmud Reza, Mohammad Zakir Hossain, Mohammad Anwar Hossain, Adittya Arefin, Ahmed Hossain.

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
