## [Decision Letter · Decision Letter 0]

24 Mar 2021

PONE-D-21-05580

National trends in the willingness to vaccinate against COVID-19 among Bangladeshi adults: Understanding the strategies to optimize vaccination coverage

PLOS ONE

Dear Dr. Hossain,

Thank you for submitting your manuscript to PLOS ONE. After careful consideration, we feel that it has merit but does not fully meet PLOS ONE’s publication criteria as it currently stands. Therefore, we invite you to submit a revised version of the manuscript that addresses the points raised during the review process.

The paper has been reviewed by two independent reviewers and they are concerned about the arguments that was made to justify the findings. There were inconsistencies in interpretation of findings and the conclusion is exaggerated. 

We look forward to receiving your revised manuscript.

Kind regards,

Enamul Kabir

Academic Editor

PLOS ONE

Journal Requirements:

3. Please provide additional details regarding participant consent.

In the ethics statement in the Methods and online submission information, you have specify that verbal consent was taken from some participants . If consent was verbal/oral, please specify:

a) whether the ethics committee approved the verbal/oral consent procedure,

b) why written consent could not be obtained, and

c) how verbal/oral consent was recorded. If your study included minors, please state whether you obtained consent from parents or guardians in these cases. If the need for consent was waived by the ethics committee, please include this information.

4. Please include additional information regarding the survey or questionnaire used in the study and ensure that you have provided sufficient details that others could replicate the analyses. For instance, if you developed a questionnaire as part of this study and it is not under a copyright more restrictive than CC-BY, please include a copy, in both the original language and English, as Supporting Information.

In the Methods section, please provide additional information regarding the questionnaire development process, including the theories or frameworks which were employed.

6. We note that Supplementary Table 1 in your submission contain map images which may be copyrighted.

We require you to either (a) present written permission from the copyright holder to publish this figure specifically under the CC BY 4.0 license, or (b) remove the figure from your submission:

a. You may seek permission from the original copyright holder of Supplementary Table 1 to publish the content specifically under the CC BY 4.0 license. 

b. If you are unable to obtain permission from the original copyright holder to publish this figure under the CC BY 4.0 license or if the copyright holder’s requirements are incompatible with the CC BY 4.0 license, please either i) remove the figure or ii) supply a replacement figure that complies with the CC BY 4.0 license. Please check copyright information on all replacement figures and update the figure caption with source information. If applicable, please specify in the figure caption text when a figure is similar but not identical to the original image and is therefore for illustrative purposes only.

Reviewers' comments:

Reviewer's Responses to Questions

**Comments to the Author**

1. Is the manuscript technically sound, and do the data support the conclusions?

Reviewer #1: Yes

Reviewer #2: Partly

2. Has the statistical analysis been performed appropriately and rigorously? 

Reviewer #1: Yes

Reviewer #2: Yes

3. Have the authors made all data underlying the findings in their manuscript fully available?

Reviewer #1: Yes

Reviewer #2: Yes

4. Is the manuscript presented in an intelligible fashion and written in standard English?

Reviewer #1: Yes

Reviewer #2: No

5. Review Comments to the Author

Reviewer #1: This study assesses National trends in the willingness to vaccinate against COVID-19 among Bangladeshi adults: Understanding the strategies to optimize vaccination coverage. This is an interesting study and indicated the common public problems of the residents in Bangeladish.

Minor comments

• You have assessed the proportion of residents who are willing to vaccinate against COVID-19, not a trend. Therefore; better to revise your title accordingly.

• As a researcher, the term “methodology” is a broad science, you did not address with this single research. Therefore, replace with the term “methods and materials”.

• In methodology, the first paragraph is too long. Therefore, either minimize it to 6-8 lines or split it into certain paragraphs.

• Please feed the source and study populations of this study.

• How did you manage more than one household heads found in one compound (such as rent)?

• Is there any plan of revisit for the households closed during data collection?

• Is it possible to calculate wealth index (PCA) rather than monthly income?

Reviewer #2: Thank you for the opportunity to review the paper. The study offered to investigate an important section in literature, and I believe it would contribute to the literature with revisions.

1. Literature on Bangladesh’s vaccination coverage is missing. There are studies that reported people’s willingness for vaccination such as: https://doi.org/10.1016/S0140-6736(20)31558-0. These need substantial build up in the Introduction.

2. There was no build up towards literature expect for a statement on “limited data on vaccine acceptance”. There should be substantial motivation to build up literature gap and address the objectives.

3. The literature ignored an important element of vaccine acceptance research: psychology and cognitive bias. There are literature on that such as: https://www.nature.com/articles/s41467-020-20226-9, https://apps.who.int/iris/rest/bitstreams/1320080/retrieve.

4. Study design and participants: Any justification for sample size ratio 500:800 between six districts and Dhaka/Chattogram?

5. Study design and participants: Why was design effect considered as 2? How is the likelihood 50%? Why was total sample size collected based on convenience and pre-determined the number (N=4600) and then provide a justification of 80% power?

6. Study design and participants: How was sampling frame determined?

7. Study design and participants: What are the definitions of semi-urban, rural and slum? How were they differentiated (supplementary-Figure 2)?

8. Study design and participants: How were the categories for chronic disease determined? A (at least) reference and justification for choosing 5 specific disease groups need to be detailed.

9. Results: “The proportions of vaccine acceptance were 68.7% and 77.4% for respondents with and without chronic diseases, respectively” – is this average proportion?

10. Results: two interpretations were reported: a) The people who had high confidence in the country's healthcare system showed a significantly lower probability of unwillingness about the COVID-19 vaccine (RRR=0.49, 95% CI=0.32-0.76) compared to people who had low confidence with the country's healthcare system. b) It appears that the proportions of vaccine acceptance by levels of self-reported compliance with healthcare rules were approximately similar.

Are they not opposite?

11. Discussion: “The study revealed geographical location, occupation, marital status, and presence of chronic condition as the significant associated factors for vaccine acceptance among the Bangladeshi population.” – This excludes education, and confidence in healthcare system, found significant in the multinomial regression.

12. While discussing the geographical differences (slum/rural vs urban), why would reference from US be relevant? Should this not be more reasonable to compare across Southeast Asia or Indian Subcontinent?

13. Relation between household autonomy and vaccine willingness is unclear. The reference (40) does not provide any such insights.

14. Conclusion seems to include overstatements that were beyond the scope of the results.

Minor:

1. Recheck the spelling of ‘Chattogram’. The government gazette changed the spelling on 10 September 2018.

2. Page 13- “8 men and eight women” – please be consistent with numbering.

6. PLOS authors have the option to publish the peer review history of their article (what does this mean?). If published, this will include your full peer review and any attached files.

Reviewer #1: No

Reviewer #2: No

---

## [Author Response · Author response to Decision Letter 0]

26 Mar 2021

March 26, 2021

Enamul Kabir

Academic Editor

PLOS ONE

PONE-D-21-05580

National trends in the willingness to vaccinate against COVID-19 among Bangladeshi adults: Understanding the strategies to optimize vaccination coverage.

Dear Professor Enamul Kabir,

Thank you very much for your editorial suggestions and the reviewers’ comments. They were accommodating. Please find enclosed an itemized list of responses along with the revised manuscript. 

In our response to the reviewer, we used regular font for the comments/questions by the referees and regular, bold font for our responses, which are shown immediately following the questions/comments.

Thank you once again for the opportunity to submit a revised manuscript.

Ahmed Hossain, PhD

Professor, Department of Public Health

Director, Global Health Institute.

North South University.

Table of Contents

EDITORIAL SUGGESTION 1

REVIEWER 1 4

REVIEWER #2: 6

Editorial suggestion 

1.Please ensure that your manuscript meets PLOS ONE's style requirements, including those for file naming 

Authors: Thank you. We revised it accordingly. 

Authors: We revised the manuscript for language usage, spelling and grammar. 

3. Please provide additional details regarding participant consent.

In the ethics statement in the Methods and online submission information, you have specify that verbal consent was taken from some participants . If consent was verbal/oral, please specify:

a) whether the ethics committee approved the verbal/oral consent procedure,

b) why written consent could not be obtained, and

c) how verbal/oral consent was recorded. If your study included minors, please state whether you obtained consent from parents or guardians in these cases. If the need for consent was waived by the ethics committee, please include this information.

 Authors: It was not an online submission information. The following statements were provided regarding participant consent. 

Verbal consents were taken for the study as the participants expressed conviction in signing or giving fingerprints on any paper, and also, many of the people from slums were analphabetic. They were reassured that all the information collected would be kept strictly confidential and would not be used for anything other than research purposes. However, they were provided with a consent paper with detailed contact information of the research investigators for any future query. The Institutional Review Board at North South University, Bangladesh approved the study (2020/OR-NSU/IRB/1003). We included the STROBE statement by completing STROBE Checklist.

4. Please include additional information regarding the survey or questionnaire used in the study and ensure that you have provided sufficient details that others could replicate the analyses. For instance, if you developed a questionnaire as part of this study and it is not under a copyright more restrictive than CC-BY, please include a copy, in both the original language and English, as Supporting Information.

In the Methods section, please provide additional information regarding the questionnaire development process, including the theories or frameworks which were employed.

 Authors: The questionnaire, R scripts, and data are available at https://osf.io/byqzv/.

 Authors: Thanks. Included. 

6. We note that Supplementary Table 1 in your submission contain map images which may be copyrighted.

Authors: The map is not copyrighted. The following r code was used to draw the map. Thanks. 

library(tidyverse)

library(ggplot2)

library(readr)

library(maps)

library(viridis)

library(maptools)

library(mapdata)

library(RColorBrewer)

map("worldHires", "Bangladesh", col="deepskyblue3",fill=F, lwd=2, mar=c(0,0,0,0))

Reviewer 1 

This study assesses National trends in the willingness to vaccinate against COVID-19 among Bangladeshi adults: Understanding the strategies to optimize vaccination coverage. This is an interesting study and indicated the common public problems of the residents in Bangladesh. 

Minor comments

R1: You have assessed the proportion of residents who are willing to vaccinate against COVID-19, not a trend. Therefore; better to revise your title accordingly. 

Authors: Thank you very much for your thoughtful comment. We changed the title to “Willingness to vaccinate against COVID-19 among Bangladeshi adults: Understanding the strategies to optimize vaccination coverage”.

R1: As a researcher, the term “methodology” is a broad science, you did not address with this single research. Therefore, replace with the term “methods and materials”.

Authors: Thanks again. We changed it to “Methods and materials”

R1: In methodology, the first paragraph is too long. Therefore, either minimize it to 6-8 lines or split it into certain paragraphs. 

Authors: Thanks. We split the paragraph into two paragraphs. 

R1: Please feed the source and study populations of this study. 

Authors: We described the study population in the section of study design and participants’ section. 

R1: How did you manage more than one household heads found in one compound (such as rent)?

Authors: Many thanks for your thoughtful comment. The first household head who came forward to participate in the study from a house was interviewed. But we did not include two members from same household. 

R1: Is there any plan of revisit for the households closed during data collection?

Authors: Thank you. We did not plan for revisit if the households were found closed or household adults were absent during the interview. We mentioned our response rate was 79.3%. 

R1:Is it possible to calculate wealth index (PCA) rather than monthly income?

Authors: In our survey, we asked about household monthly income as well as household characteristics. Unfortunately, many respondents did not respond to questions about household characteristics, so the wealth index calculated using principal component analysis (PCA) did not provide a wealth score for many households.

 

Reviewer #2: 

Thank you for the opportunity to review the paper. The study offered to investigate an important section in literature, and I believe it would contribute to the literature with revisions.

1. Literature on Bangladesh’s vaccination coverage is missing. There are studies that reported people’s willingness for vaccination such as: https://doi.org/10.1016/S0140-6736(20)31558-0. These need substantial build up in the Introduction.

Authors: Many thanks for your comment and the reference. This reference [29] was included in the manuscript. We also included a section to understand Bangladesh’s vaccination coverage till March 25, 2021. 

2. There was no build up towards literature expect for a statement on “limited data on vaccine acceptance”. There should be substantial motivation to build up literature gap and address the objectives.

Authors: A paragraph is included to build up towards motivation of the study. The paragraph is in the following: 

Although evidence on encouraging vaccination in general is useful in the context of the current pandemic, COVID-19 vaccine acceptance and uptake pose an enormous challenge [18]. Sentiment that sows doubt and mistrust, as well as the viral dissemination of misinformation, are both leading to a community of vaccine reluctance [19]. Understanding the changing trends of vaccine acceptance over time, which can serve as an early warning system for taking necessary steps to prevent decreases in vaccine trust and acceptance, requires a standard measure of confidence and a baseline for comparison. The study offers a valuable baseline of confidence levels to assess willingness to vaccinate in the context of the COVID-19 pandemic and to help identify where further trust building is required to maximize adoption of new life-saving vaccines.

3. The literature ignored an important element of vaccine acceptance research: psychology and cognitive bias. There are literatures on that such as: https://www.nature.com/articles/s41467-020-20226-9, https://apps.who.int/iris/rest/bitstreams/1320080/retrieve.

Authors: Many thanks for these two wonderful references. We have included both references in the manuscript.

4. Study design and participants: Any justification for sample size ratio 500:800 between six districts and Dhaka/Chattogram?

Authors: For each district (assumed equal cluster size), the sample allocation was kept at 500, and for Dhaka and Chattogram, it was kept at 800. It was predetermined because it would give the largest sample size possible with a response rate of 75% from each district. We chose Dhaka and Chattogram for city and slum sampling, and both the city and slum populations were stratified. Also, Dhaka and Chattogram have greater populations than the other six districts. So, we planned to sample Dhaka and Chattogram more than other districts. As a result, we intend to recruit 4600 people for the study.

5. Study design and participants: Why was design effect considered as 2? How is the likelihood 50%? Why was total sample size collected based on convenience and pre-determined the number (N=4600) and then provide a justification of 80% power?

Authors: According to our understanding, a design effect (DEFF) is an adjustment made to find a survey sample size due to a sampling method (e.g. cluster sampling) that results a large sample sizes. A DEFF of 2 indicates that the variance is twice what we would expect from random sampling. It also implies that, in order to optimize the sample size, we can use twice the sample size if we use cluster sampling. 

We did not consider likelihood 50%. We mentioned an assumption of 50% of willingness to vaccinate because it will provide the maximum sample size needed for a survey. 

The survey sampling has three stages. In one stage, districts selection was done conveniently because we did have opportunities of data collection in these districts. However, the household selection was done by systematic sampling approach. 

Again, we did not consider 80% power. We mentioned if anyone consider a minimum power of 80% then our sample size is higher than the calculated sample size. Please find following the r code from epiR package and it shows with 200 samples per cluster we need 8 clusters and a minimum sample size is needed 1469 samples. 

6. Study design and participants: How was sampling frame determined?

Authors: It was a three-stage sampling method, as we discussed earlier. We didn't have a sampling frame to work with in the survey. Such type of sampling approach was given in World Health Organization 2008, Training for mid-level managers (MLM)7. The EPI coverage survey. 

https://www.who.int/immunization/documents/MLM_module7.pdf

7. Study design and participants: What are the definitions of semi-urban, rural and slum? How were they differentiated (supplementary-Figure 2)?

Authors: The participants reported it, and the interviewer kept track of it. The interviewer went to a village, and if the respondents had lived there for the previous year, the participant's residence was listed as rural. Similarly, respondents from slums were interviewed within slum areas, and upazila participants were considered semi-urban.

8. Study design and participants: How were the categories for chronic disease determined? A (at least) reference and justification for choosing 5 specific disease groups need to be detailed.

Authors: Thank you for your suggestion. We detailed it in the manuscript and the six common diseases were considered because the respondents can report the diseases conveniently. 

To collect data for an underlying chronic condition, participants were asked if they were suffering from any of the six common chronic conditions: diabetes, hypertension, chronic kidney disease (CKD), chronic respiratory disease (CRD), chronic heart disease, and cancer. The response was recorded as ‘Yes/No’. Further, the participants were asked if they were infected by the SARS-CoV-2 virus, and the responses were recorded as ‘Yes/No’. The participants self-reported their chronic illnesses and infections in the survey.

9. Results: “The proportions of vaccine acceptance were 68.7% and 77.4% for respondents with and without chronic diseases, respectively” – is this average proportion?

Authors: For clarity we revised the sentence as “The proportion of vaccine acceptance was 68.7% who reported any of the chronic diseases, while the proportion was 77.4% for respondents without any chronic illness.”

10. Results: two interpretations were reported: a) The people who had high confidence in the country's healthcare system showed a significantly lower probability of unwillingness about the COVID-19 vaccine (RRR=0.49, 95% CI=0.32-0.76) compared to people who had low confidence with the country's healthcare system. b) It appears that the proportions of vaccine acceptance by levels of self-reported compliance with healthcare rules were approximately similar.

Are they not opposite?

Authors: (a) is correct. It means those who had high confidence in the country’s health care system showed more willingness to vaccinate (i.e., low probability of unwillingness). 

(b) is also correct. The proportions of vaccine acceptance by levels of self-reported compliance with healthcare rules (e.g., wearing masks) were approximately similar because the proportions were 75.9%, 74.2% and 70.9% for low compliant group, moderate compliant group and high compliant group, respectively.

11. Discussion: “The study revealed geographical location, occupation, marital status, and presence of chronic condition as the significant associated factors for vaccine unwillingness among the Bangladeshi population.” – This excludes education, and confidence in healthcare system, found significant in the multinomial regression.

Authors: The more educated people expressed uncertainty but did not show unwillingness. It was addressed in more detail in the later paragraph. That is why, in our statement of unwillingness, we did not include education. In the statement, we included the variable that indicated confidence in the healthcare system. 

12. While discussing the geographical differences (slum/rural vs urban), why would reference from US be relevant? Should this not be more reasonable to compare across Southeast Asia or Indian Subcontinent?

Authors: We showed both Indian and US results in the manuscript. For example, close results have been found in a similar survey in Mumbai slum in India, which also demonstrated a 20% unacceptance of vaccine for COVID-19 among the slum dwellers [32]. According to a nationwide poll conducted in the United States, rural residents had a higher rate of vaccine hesitancy [33]. 

We believe it is critical to demonstrate how the proportion of people who are willing to be vaccinated varies across the world. If the United States becomes more unwilling, there is a risk that the proportion of willingness in South and Southeast Asian countries will decrease.

13. Relation between household autonomy and vaccine willingness is unclear. The reference (40) does not provide any such insights.

Authors: The decision-making domain in a household may have an impact on the women's willingness to vaccinated. Homemakers (housewives) were the group with the highest percentage of people who refused to take the COVID-19 vaccine. The household head in Bangladeshi culture is a male, and women's lack of decision-making autonomy can be related to their unwillingness to get vaccinated against COVID-19 [40].

14. Conclusion seems to include overstatements that were beyond the scope of the results.

Authors: Many thanks for your comment. We revised the conclusion as following:

The results of the analysis illustrate the challenges in introducing a new treatment protocol or some other health initiative through a population in a developing country like Bangladesh. The findings will aid policymakers in developing successful immunization policies, implementing vaccination programs, locating and prioritizing accurate target populations to ensure easy access to COVID-19 vaccines, addressing vaccine hesitancy concerns, and increasing public interest in the vaccine. Media and communication experts will be benefitted from crafting their message targeting the right audiences effectively. Our findings highlight that the Government of Bangladesh should take effective steps to develop a tailor-made vaccine campaign strategy for rural people and slum dwellers, especially targeting farmers, day laborers, and homemakers. Communication messages must be designed to be interpreted easily for people with low literacy; thereby, these people can grow trust in the healthcare system and accept the COVID-19 vaccine. The government must take the necessary steps to ensure the convenience of vaccination in rural areas and the accessibility for the slum dwellers. Attention should also be given to individuals suffering from chronic illness with personalized health messages from healthcare professionals.

Minor:

1. Recheck the spelling of ‘Chattogram’. The government gazette changed the spelling on 10 September 2018.

Authors: Thank you very much. We changed it to “Chattogram”.

2. Page 13- “8 men and eight women” – please be consistent with numbering.

Authors: Thank you. We changed it to “Eight men and eight women”

---

## [Decision Letter · Decision Letter 1]

8 Apr 2021

Willingness to vaccinate against COVID-19 among Bangladeshi adults: Understanding the strategies to optimize vaccination coverage

PONE-D-21-05580R1

Dear Dr. Hossain,

We’re pleased to inform you that your manuscript has been judged scientifically suitable for publication and will be formally accepted for publication once it meets all outstanding technical requirements.

Kind regards,

Enamul Kabir

Academic Editor

PLOS ONE

Additional Editor Comments (optional):

Reviewers' comments:

Reviewer's Responses to Questions

**Comments to the Author**

1. If the authors have adequately addressed your comments raised in a previous round of review and you feel that this manuscript is now acceptable for publication, you may indicate that here to bypass the “Comments to the Author” section, enter your conflict of interest statement in the “Confidential to Editor” section, and submit your "Accept" recommendation.

Reviewer #1: All comments have been addressed

Reviewer #2: (No Response)

2. Is the manuscript technically sound, and do the data support the conclusions?

Reviewer #1: Yes

Reviewer #2: (No Response)

3. Has the statistical analysis been performed appropriately and rigorously? 

Reviewer #1: Yes

Reviewer #2: (No Response)

4. Have the authors made all data underlying the findings in their manuscript fully available?

Reviewer #1: Yes

Reviewer #2: (No Response)

5. Is the manuscript presented in an intelligible fashion and written in standard English?

Reviewer #1: Yes

Reviewer #2: (No Response)

6. Review Comments to the Author

Reviewer #1: (No Response)

Reviewer #2: (No Response)

7. PLOS authors have the option to publish the peer review history of their article (what does this mean?). If published, this will include your full peer review and any attached files.

Reviewer #1: No

Reviewer #2: No